# BEHAVIOR REGULARIZED OFFLINE REINFORCEMENT LEARNING

## ABSTRACT

In reinforcement learning (RL) research, it is common to assume access to direct *online* interactions with the environment. However in many real-world applications, access to the environment is limited to a fixed *offline* dataset of logged experience. In such settings, standard RL algorithms have been shown to diverge or otherwise yield poor performance. Accordingly, much recent work has suggested a number of remedies to these issues. In this work, we introduce a general framework, *behavior regularized actor critic* (BRAC), to empirically evaluate recently proposed methods as well as a number of simple baselines across a variety of offline continuous control tasks. Surprisingly, we find that many of the technical complexities introduced in recent methods are unnecessary to achieve strong performance. Additional ablations provide insights into which design choices matter most in the offline RL setting.

## 1 INTRODUCTION

Offline reinforcement learning (RL) describes the setting in which a learner has access to only a fixed dataset of experience. In contrast to online RL, additional interactions with the environment during learning are not permitted. This setting is of particular interest for applications in which deploying a policy is costly or there is a safety concern with updating the policy online (Li et al., 2015). For example, for recommendation systems (Li et al., 2011; Covington et al., 2016) or health applications (Murphy et al., 2001), deploying a new policy may only be done at a low frequency after extensive testing and evaluation. In these cases, the offline dataset is often very large, potentially encompassing years of logged experience. Nevertheless, the inability to interact with the environment directly poses a challenge to modern RL algorithms.

Issues with RL algorithms in the offline setting typically arise in cases where state and actions spaces are large or continuous, necessitating the use of function approximation. While off-policy (deep) RL algorithms such as DQN (Mnih et al., 2013), DDPG (Lillicrap et al., 2015), and SAC (Haarnoja et al., 2018) may be run directly on offline datasets to learn a policy, the performance of these algorithms has been shown to be sensitive to the experience dataset distribution, even in the online setting when using a replay buffer (Van Hasselt et al., 2018; Fu et al., 2019). Moreover, Fujimoto et al. (2018a) and Kumar et al. (2019) empirically confirm that in the offline setting, DDPG fails to learn a good policy, even when the dataset is collected by a single behavior policy, with or without noise added to the behavior policy. These failure cases are hypothesized to be caused by erroneous generalization of the state-action value function (Q-value function) learned with function approximators, as suggested by Sutton (1995); Baird (1995); Tsitsiklis & Van Roy (1997); Van Hasselt et al. (2018). To remedy this issue, two types of approaches have been proposed recently: 1) Agarwal et al. (2019) proposes to apply a random ensemble of Q-value targets to stabilize the learned Q-function, 2) Fujimoto et al. (2018a); Kumar et al. (2019); Jaques et al. (2019); Laroche & Trichelair (2017) propose to regularize the learned policy towards the behavior policy based on the intuition that unseen state-action pairs are more likely to receive overestimated Q-values. These proposed remedies have been shown to improve upon DQN or DDPG at performing policy improvement based on offline data. Still, each proposal makes several modifications to the building components of baseline off-policy RL algorithms, and each modification may be implemented in various ways. So a natural question to ask is, which of the design choices in these offline RL algorithms are necessary to achieve good performance? For example, to estimate the target Q-value when minimizing the Bellman error, Fujimoto et al. (2018a) uses a soft combination of two target Q-values, which is different from TD3

(Fujimoto et al., 2018b), where the minimum of two target Q-values is used. This soft combination is maintained by Kumar et al. (2019), while further increasing the number of Q-networks from two to four. As another example, when regularizing towards the behavior policy, Jaques et al. (2019) uses Kullback-Leibler (KL) divergence with a fixed regularization weight while Kumar et al. (2019) proposes to use Maximum Mean Discrepancy (MMD) with an adaptively trained regularization weight. Are these design choices crucial to success in offline settings? Or are they simply the result of multiple, human-directed iterations of research?

In this work, we aim at evaluating the importance of different algorithmic building components as well as comparing different design choices in offline RL approaches. We focus on behavior regularized approaches applied to continuous action domains, encompassing many of the recently demonstrated successes (Fujimoto et al., 2018a; Kumar et al., 2019). We introduce *behavior regularized actor critic* (BRAC), a general algorithmic framework which covers existing approaches while enabling us to compare the performance of different variants in a modular way. We find that many simple variants of the behavior regularized approach can yield good performance, while previously suggested sophisticated techniques such as weighted Q-ensembles and adaptive regularization weights are not crucial. Experimental ablations reveal further insights into how different design choices affect the performance and robustness of the behavior regularized approach in the offline RL setting.

## 2 BACKGROUND

### 2.1 MARKOV DECISION PROCESSES

We consider the standard fully-observed Markov Decision Process (MDP) setting (Puterman, 1990). An MDP can be represented as $\mathcal{M} = (\mathcal{S}, \mathcal{A}, P, R, \gamma)$ where $\mathcal{S}$ is the state space, $\mathcal{A}$ is the action space, $P(\cdot|s,a)$ is the transition probability distribution function, $R(s,a)$ is the reward function and $\gamma$ is the discount factor. The goal is to find a policy $\pi(\cdot|s)$ that maximizes the cumulative discounted reward starting from any state $s \in \mathcal{S}$. Let $P^\pi(\cdot|s)$ denote the induced transition distribution for policy $\pi$. For later convenience, we also introduce the notion of multi-step transition distributions as $P_t^\pi$, where $P_t^\pi(\cdot|s)$ denotes the distribution over the state space after rolling out $P^\pi$ for $t$ steps starting from state $s$. For example, $P_0^\pi(\cdot|s)$ is the Dirac delta function at $s$ and $P_1^\pi(\cdot|s) = P^\pi(\cdot|s)$. We use $R^\pi(s)$ to denote the expected reward at state $s$ when following policy $\pi$, i.e. $R^\pi(s) = \mathbb{E}_{a\sim\pi(\cdot|s)}[R(s,a)]$. The state value function (a.k.a. value function) is defined by $V^\pi(s) = \sum_{t=0}^\infty \gamma^t \mathbb{E}_{s_t \sim P_t^\pi(s)}[R^\pi(s_t)]$. The action-value function (a.k.a. Q-function) can be written as $Q^\pi(s,a) = R(s,a) + \gamma\mathbb{E}_{s'\sim P(\cdot|s,a)}[V^\pi(s')]$. The optimal policy is defined as the policy $\pi^*$ that maximizes $V^{\pi^*}(s)$ at all states $s \in \mathcal{S}$. In the commonly used actor critic paradigm, one optimizes a policy $\pi_\theta(\cdot|s)$ by alternatively learning a Q-value function $Q_\psi$ to minimize Bellman errors over single step transitions $(s,a,r,s')$, $\mathbb{E}_{a'\sim\pi_\theta(\cdot|s')}\left[\left(r + \gamma\bar{Q}(s',a') - Q_\psi(s,a)\right)^2\right]$, where $\bar{Q}$ denotes a target Q function; e.g., it is common to use a slowly-updated target parameter set $\psi'$ to determine the target Q function as $Q_{\psi'}(s',a')$. Then, the policy is updated to maximize the Q-values, $\mathbb{E}_{a\sim\pi(\cdot|s)}[Q_\psi(s,a)]$.

### 2.2 OFFLINE REINFORCEMENT LEARNING

Offline RL (also known as batch RL (Lange et al., 2012)) considers the problem of learning a policy $\pi$ from a fixed dataset $\mathcal{D}$ consisting of single-step transitions $(s,a,r,s')$. Slightly abusing the notion of "behavior", we define the behavior policy $\pi_b(a|s)$ as the conditional distribution $p(a|s)$ observed in the dataset distribution $\mathcal{D}$. Under this definition, such a behavior policy $\pi_b$ is always well-defined even if the dataset was collected by multiple, distinct behavior policies. Because we do not assume direct access to $\pi_b$, it is common in previous work to approximate this behavior policy with max-likelihood over $\mathcal{D}$:

$$\hat{\pi}_b := \underset{\hat{\pi}}{\operatorname{argmax}} \, \mathbb{E}_{(s,a,r,s')\sim\mathcal{D}}\left[\log\hat{\pi}(a|s)\right]. \tag{1}$$

We denote the learned policy as $\hat{\pi}_b$ and refer to it as the "cloned policy" to distinguish it from the true behavior policy.

In this work, we focus on the offline RL problem for complex continuous domains. We briefly review two recently proposed approaches, BEAR (Kumar et al., 2019) and BCQ (Fujimoto et al., 2018a).

**BEAR** Motivated by the hypothesis that deep RL algorithms generalize poorly to actions outside the support of the behavior policy, Kumar et al. (2019) propose BEAR, which learns a policy to maximize Q-values while penalizing it from diverging from behavior policy support. BEAR measures divergence from the behavior policy using kernel MMD (Gretton et al., 2007):

$$\text{MMD}_k^2(\pi(\cdot|s), \pi_b(\cdot|s)) = \mathbb{E}_{x,x'\sim\pi(\cdot|s)}[K(x,x')] - 2\mathbb{E}_{\substack{x\sim\pi(\cdot|s)\\y\sim\pi_b(\cdot|s)}}[K(x,y)] + \mathbb{E}_{y,y'\sim\pi_b(\cdot|s)}[K(y,y')] , \quad (2)$$

where $K$ is a kernel function. Furthermore, to avoid overestimation in the Q-values, the target Q-value function $\bar{Q}$ is calculated as,

$$\bar{Q}(s', a') := 0.75 \cdot \min_{j=1,\dots,k} Q_{\psi'_j}(s', a') + 0.25 \cdot \max_{j=1,\dots,k} Q_{\psi'_j}(s', a'), \quad (3)$$

where $\psi'_j$ is denotes a soft-updated ensemble of target Q functions. In BEAR's implementation, this ensemble is of size $k = 4$. BEAR also penalizes target Q-values by an ensemble variance term. However, their empirical results show that there is no clear benefit to doing so, thus we omit this term.

**BCQ** BCQ enforces $\pi$ to be close to $\pi_b$ with a specific parameterization of $\pi$:

$$\pi_\theta(a|s) := \underset{a_i+\xi_\theta(s,a_i)}{\text{argmax}} Q_\psi(s, a_i + \xi_\theta(s, a_i)) \text{ for } a_i \sim \pi_b(a|s), \ i = 1, \dots, N, \quad (4)$$

where $\xi_\theta$ is a function approximator with bounded output in $[-\Phi, \Phi]$ where $\Phi$ is a hyperparameter. $N$ is an additional hyperparameter used during evaluation to compute $\pi_\theta$ and during training for Q-value updates. The target Q-value function $\bar{Q}$ is calculated as in Equation 3 but with $k = 2$.

## 3 BEHAVIOR REGULARIZED ACTOR CRITIC

Encouraging the learned policy to be close to the behavior policy is a common theme in previous approaches to offline RL. To evaluate the effect of different behavior policy regularizers, we introduce *behavior regularized actor critic* (BRAC), an algorithmic framework which generalizes existing approaches while providing more implementation options.

There are two common ways to incorporate regularization to a specific policy: through a penalty in the value function or as a penalty solely on the policy. We begin by introducing the former, **value penalty (vp)**. Similar to SAC (Haarnoja et al., 2018) which adds an entropy term to the target Q-value calculation, we add a term to the target Q-value calculation that regularizes the learned policy $\pi$ towards the behavior policy $\pi_b$. Specifically, we define the penalized value function as

$$V_D^\pi(s) = \sum_{t=0}^\infty \gamma^t \mathbb{E}_{s_t \sim P_t^\pi(s)}[R^\pi(s_t) - \alpha D(\pi(\cdot|s_t), \pi_b(\cdot|s_t))] , \quad (5)$$

where $D$ is a divergence function between distributions over actions (e.g., MMD or KL divergence). Following the typical actor critic framework, the Q-value objective is given by,

$$\min_{Q_\psi} \mathbb{E}_{\substack{(s,a,r,s')\sim\mathcal{D}\\a'\sim\pi_\theta(\cdot|s')}}\left[\left(r + \gamma\left(\bar{Q}(s', a') - \alpha\hat{D}(\pi_\theta(\cdot|s'), \pi_b(\cdot|s'))\right) - Q_\psi(s, a)\right)^2\right], \quad (6)$$

where $\bar{Q}$ again denotes a target Q function and $\hat{D}$ denotes a sample-based estimate of the divergence function $D$. The policy learning objective can be written as,

$$\max_{\pi_\theta} \mathbb{E}_{(s,a,r,s')\sim\mathcal{D}}\left[\mathbb{E}_{a''\sim\pi_\theta(\cdot|s)}[Q_\psi(s, a'')] - \alpha\hat{D}(\pi_\theta(\cdot|s), \pi_b(\cdot|s))\right]. \quad (7)$$

Accordingly, one performs alternating gradient updates based on (6) and (7). This algorithm is equivalent to SAC when using a single-sample estimate of the entropy for $\hat{D}$; i.e., $\hat{D}(\pi_\theta(\cdot|s'), \pi_b(\cdot|s')) := \log \pi(a'|s')$ for $a' \sim \pi(\cdot|s')$.

The second way to add the regularizer is to only regularize the policy during policy optimization. That is, we use the same objectives in Equations 6 and 7, but use $\alpha = 0$ in the Q update while using a non-zero $\alpha$ in the policy update. We call this variant **policy regularization (pr)**. This proposal is similar to the regularization employed in A3C (Mnih et al., 2016), if one uses the entropy of $\pi_\theta$ to compute $\hat{D}$.

In addition to the choice of value penalty or policy regularization, the choice of $D$ and how to perform sample estimation of $\hat{D}$ is a key design choice of BRAC:

**Kernel MMD** We can compute a sample based estimate of kernel MMD (Equation 2) by drawing samples from both $\pi_\theta$ and $\pi_b$. Because we do not have access to multiple samples from $\pi_b$, this requires a pre-estimated cloned policy $\hat{\pi}_b$.

**KL Divergence** With KL Divergence, the behavior regularizer can be written as

$$D_{\text{KL}}\left(\pi_\theta(\cdot|s), \pi_b(\cdot|s)\right) = \mathbb{E}_{a \sim \pi_\theta(\cdot|s)}\left[\log \pi_\theta(a|s) - \log \pi_b(a|s)\right] .$$

Directly estimating $D_{\text{KL}}$ via samples requires having access to the density of both $\pi_\theta$ and $\pi_b$; as in MMD, the cloned $\hat{\pi}_b$ can be used in place of $\pi_b$. Alternatively, we can avoid estimating $\pi_b$ explicitly, by using the dual form of the KL-divergence. Specifically, any $f$-divergence (Csiszár, 1964) has a dual form (Nowozin et al., 2016) given by,

$$D_f(p, q) = \mathbb{E}_{x \sim p}\left[f\left(q(x)/p(x)\right)\right] = \max_{g:\mathcal{X} \mapsto \text{dom}(f^*)} \mathbb{E}_{x \sim q}\left[g(x)\right] - \mathbb{E}_{x \sim p}\left[f^*(g(x))\right] ,$$

where $f^*$ is the Fenchel dual of $f$. In this case, one no longer needs to estimate a cloned policy $\hat{\pi}_b$ but instead needs to learn a discriminator function $g$ with minimax optimization as in Nowozin et al. (2016). This sample based dual estimation can be applied to any $f$-divergence. In the case of a KL-divergence, $f(x) = -\log x$ and $f^*(t) = -\log(-t) - 1$.

**Wasserstein Distance** One may also use the Wassertein distance as the divergence $D$. For sample-based estimation, one may use its dual form,

$$W(p, q) = \sup_{g:||g||_L \leq 1} \mathbb{E}_{x \sim p}\left[g(x)\right] - \mathbb{E}_{x \sim q}\left[g(x)\right]$$

and maintain a discriminator $g$ as in Gulrajani et al. (2017).

Now we discuss how existing approaches can be instantiated under the framework of BRAC.

**BEAR** To re-create BEAR with BRAC, one uses policy regularization with the sample-based kernel MMD for $\hat{D}$ and uses a min-max ensemble estimate for $\bar{Q}$ (Equation 3). Furthermore, BEAR adaptively trains the regularization weight $\alpha$ as a Lagriagian multiplier: it sets a threshold $\epsilon > 0$ for the kernel MMD distance and increases $\alpha$ if the current average divergence is above the threshold and decreases $\alpha$ if below the threshold.

**BCQ** The BCQ algorithm does not use any regularizers (i.e. $\alpha = 0$ for both value and policy objectives). Still, the algorithm may be realized by BRAC if one restricts the policy optimization in Equation 7 to be over parameterized policies based on Equation 4.

**KL-Control** There has been a rich set of work which investigates regularizing the learned policy through KL-divergence with respect to another policy, e.g. Abdolmaleki et al. (2018); Kakade (2002); Peters et al. (2010); Schulman et al. (2015); Nachum et al. (2017). Notably, Jaques et al. (2019) apply this idea to offline RL in discrete action domains by introducing a KL value penalty in the Q-value definition. It is clear that BRAC can realize this algorithm as well.

To summarize, one can instantiate the behavior regularized actor critic framework with different design choices, including how to estimate the target Q value, which divergence to use, whether to learn $\alpha$ adaptively, whether to use a value penalty in the Q function objective (6) or just use policy regularization in (7) and so on. In the next section, we empirically evaluate a set of these different design choices to provide insights into what actually matters when approaching the offline RL problem.

## 4 EXPERIMENTS

The BRAC framework encompasses several previously proposed methods depending on specific design choices (e.g., whether to use value penalty or policy regularization, how to compute the target Q-value, and how to impose the behavior regularization). For a practitioner, key questions are: How should these design choices be made? Which variations among these different algorithms actually matter? To answer these questions, we perform a systematic evaluation of BRAC under different design choices.

Following Kumar et al. (2019), we evaluate performance on four Mujoco (Todorov et al., 2012) continuous control environments in OpenAI Gym (Brockman et al., 2016): Ant-v2, HalfCheetah-v2, Hopper-v2, and Walker2d-v2. In many real-world applications of RL, one has logged data from sub-optimal policies (e.g., robotic control and recommendation systems). To simulate this scenario, we collect the offline dataset with a sub-optimal policy perturbed by additional noise. To obtain a partially trained policy, we train a policy with SAC and online interactions until the policy performance achieves a performance threshold $(1000, 4000, 1000, 1000$ for Ant-v2, HalfCheetah-v2, Hopper-v2, Walker2d-v2, respectively, similar to the protocol established by Kumar et al. (2019)). Then, we perturb the partially trained policy with noise (Gaussian noise or $\epsilon$-greedy at different levels) to simulate different exploration strategies resulting in five noisy behavior policies. We collect 1 million transitions according to each behavior policy resulting in five datasets for each environment (see Appendix for implementation details). We evaluate offline RL algorithms by training on these fixed datasets and evaluating the learned policies on the real environments.

In preliminary experiments, we found that policy learning rate and regularization strength have a significant effect on performance. As a result, for each variant of BRAC and each environment, we do a grid search over policy learning rate and regularization strength. For policy learning rate, we search over six values, ranging from $3 \cdot 10^6$ to $0.001$. The regularization strength is controlled differently in different algorithms. In the simplest case, the regularization weight $\alpha$ is fixed; in BEAR the regularization weight is adaptively trained with dual gradient ascent based on a divergence constraint $\epsilon$ that is tuned as a hyperparameter; in BCQ the corresponding tuning is for the perturbation range $\Phi$. For each of these options, we search over five values (see Appendix for details). For existing algorithms such as BEAR and BCQ, the reported hyperparameters in their papers (Kumar et al., 2019; Fujimoto et al., 2018a) are included in this search range, We select the best hyperparameters according to the average performance over all five datasets.

Currently, BEAR (Kumar et al., 2019) provides state-of-the-art performance on these tasks, so to understand the effect of variations under our BRAC framework, we start by implementing BEAR in BRAC and run a series of comparisons by varying different design choices: adaptive vs. fixed regularization, different ensembles for estimating target Q-values, value penalty vs. policy regularization and divergence choice for the regularizer. We then evaluate BCQ, which has a different design in the BRAC framework, and compare it to other BRAC variants as well as several baseline algorithms.

## 4.1 FIXED V.S. ADAPTIVE REGULARIZATION WEIGHTS

In BEAR, regularization is controlled by a threshold $\epsilon$, which is used for adaptively training the Lagrangian multiplier $\alpha$, whereas typically (e.g., in KL-control) one uses a fixed $\alpha$. In our initial experiments with BEAR, we found that when using the recommended value of $\epsilon$, the learned value of $\alpha$ consistently increased during training, implying that the MMD constraint between $\pi_\theta$ and $\pi_b$ was almost never satisfied. This suggests that BEAR is effectively performing policy regularization with a large $\alpha$ rather than constrained optimization. This led us to question if adaptively training $\alpha$ is better than using a fixed $\alpha$. To investigate this question, we evaluate the performance of both approaches (with appropriate hyperparameter tuning for each, over either $\alpha$ or $\epsilon$) in Figure 1. On most datasets, both approaches learn a policy that is much better than the *partially trained policy*[1], although we do observe a consistent modest advantage when using a fixed $\alpha$. Because using a fixed $\alpha$ is simpler and performs better than adaptive training, we use this approach in subsequent experiments.

## 4.2 ENSEMBLE FOR TARGET Q-VALUES

Another important design choice in BRAC is how to compute the target Q-value, and specifically, whether one should use the sophisticated ensemble strategies employed by BEAR and BCQ. Both BEAR and BCQ use a weighted mixture of the minimum and maximum among multiple learned Q-functions (compared to TD3 which simply uses the minimum of two). BEAR further increases the number of Q-functions from 2 to 4. To investigate these design choices, we first experiment with different number of Q-functions $k = \{1, 2, 4\}$. Results are shown in Figure 2. Fujimoto et al. (2018b) show that using two Q-functions provides significant improvements in online RL; similarly, we find that using $k = 1$ sometimes fails to learn a good policy (e.g., in Walker2d) in the offline setting.

---

[1]The partially trained policy is the policy used to collect data *without* injected noise. The true *behavior policy* and *behavior cloning* will usually get worse performance due to injected noise when collecting the data.

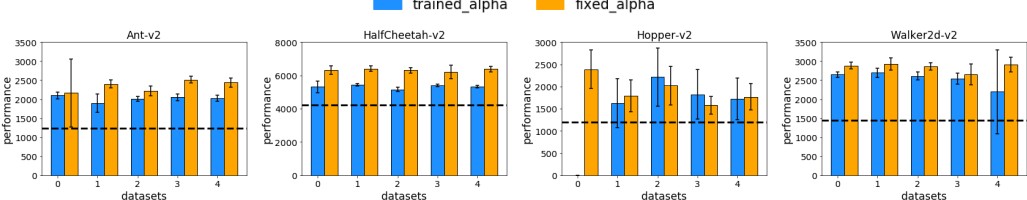

Figure 1: Comparing fixed $\alpha$ with adaptively trained $\alpha$. Black dashed lines are the performance of the partially trained policies (distinct from the behavior policies which have injected noise). We report the mean over the last 10 evaluation points (during training) averaged over 5 different random seeds. Each evaluation point is the return averaged over 20 episodes. We report the performance as 0 if it is negative.

Using $k = 4$ has a small advantage compared to $k = 2$ except in Hopper. Both $k = 2$ and $k = 4$ significantly improve over the partially trained policy baseline. In general, increasing the value of $k$ in ensemble will lead to more stable or better performance, but requires more computation cost. On these domains we found that $k = 4$ only gives marginal improvement over $k = 2$, so we use $k = 2$ in our remaining experiments.

Regarding whether using a weighed mixture of Q-values or the minimum, we compare these two options under $k = 2$. Results are shown in Figure 3. We find that taking the minimum performs slightly better than taking a mixture except in Hopper, and both successfully outperform the partially trained policy in all cases. Due to the simplicity and strong performance of taking the minimum of two Q-functions, we use this approach in subsequent experiments.

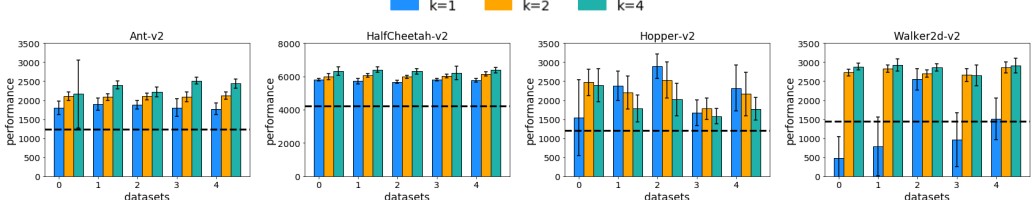

Figure 2: Comparing different number of Q-functions for target Q-value ensemble. We use a weighted mixture to compute the target value for all of these variants. As expected, we find that using an ensemble ($k > 1$) is better than using a single Q-function.

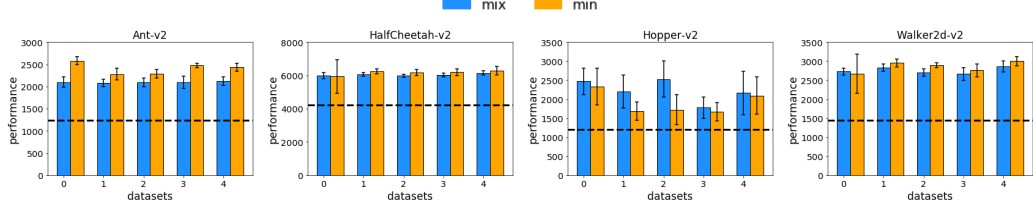

Figure 3: Comparing taking the minimum v.s. a weighted mixture in Q-value ensemble. We find that simply taking the minimum is usually slightly better, except in Hopper-v2.

### 4.3 VALUE PENALTY OR POLICY REGULARIZATION

So far, we have evaluated variations in regularization weights and ensemble of Q-values. We found that the technical complexity introduced in recent works is not always necessary to achieve state-of-the-art performance. With these simplifications, we now evaluate a major variation of design choices in BRAC — using *value penalty* or *policy regularization*. We follow our simplified version of BEAR: MMD policy regularization, fixed $\alpha$, and computation of target Q-values based on the minimum of a $k = 2$ ensemble. We compare this instantiation of BRAC to its value penalty version, with results shown in Figure 4. While both variants outperform the partially trained policy, we find that value penalty performs slightly better than policy regularization in most cases. We consistently observed this advantage with other divergence choices (see Appendix Figure 8 for a full comparison).

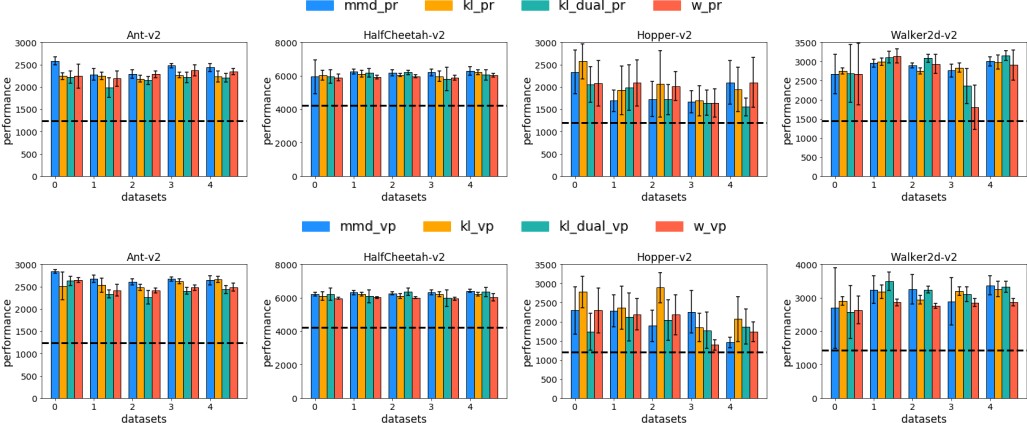

Figure 4: Comparing policy regularization (pr) v.s. value penalty (vp) with MMD. The use of value penalty is usually slightly better.

## 4.4 DIVERGENCES FOR REGULARIZATION

We evaluated four choices of divergences used as the regularizer $D$: (a) MMD (as in BEAR), (b) KL in the primal form with estimated behavior policy (as in KL-control), and (c) KL and (d) Wasserstein in their dual forms without estimating a behavior policy. As shown in Figure 5, we do not find any specific divergence performing consistently better or worse than the others. All variants are able to learn a policy that significantly improves over the behavior policy in all cases.

In contrast, Kumar et al. (2019) argue that sampled MMD is superior to KL based on the idea that it is better to regularize the support of the learned policy distribution to be within the support of the behavior policy rather than forcing the two distributions to be similar. While conceptually reasonable, we do not find support for that argument in our experiments: (i) we find that KL and Wasserstein can perform similarly well to MMD even though they are not designed for support matching; (ii) we briefly tried divergences that are explicitly designed for support matching (the relaxed KL and relaxed Wasserstein distances proposed by Wu et al. (2019)), but did not observe a clear benefit to the additional complexity. We conjecture that this is because even if one uses noisy or multiple behavior policies to collect data, the noise is reflected more in the diversity of states rather than the diversity of actions on a single state (due to the nature of environment dynamics). However, we expect this support matching vs. distribution matching distinction may matter in other scenarios such as smaller state spaces or contextual bandits, which is a potential direction for future work.

Figure 5: Comparing different divergences under both policy regularization (top row) and value penalty (bottom row). All variants yield similar performance, which is significantly better than the partially trained policy.

## 4.5 COMPARISON TO BCQ AND OTHER BASELINES

We now compare one of our best performing algorithms so far, kl_vp (value penalty with KL divergence in the primal form), to BCQ, BEAR, and two other baselines: vanilla SAC (which uses adaptive entropy regularization) and behavior cloning. Figure 6 shows the comparison. We find that vanilla SAC only works in the HalfCheetah environment and fails in the other three environments. Behavior cloning never learns a better policy than the partially trained policy used to collect the data. Although BCQ consistently learns a policy that is better than the partially trained policy, its performance is always clearly worse than kl_vp (and other variants whose performance is similar to kl_vp, according to our previous experiments). We conclude that BCQ is less favorable than explicitly

using a divergence for behavior regularization (BEAR and kl_vp). Although, tuning additional hyperparameters beyond Φ for BCQ may improve performance.

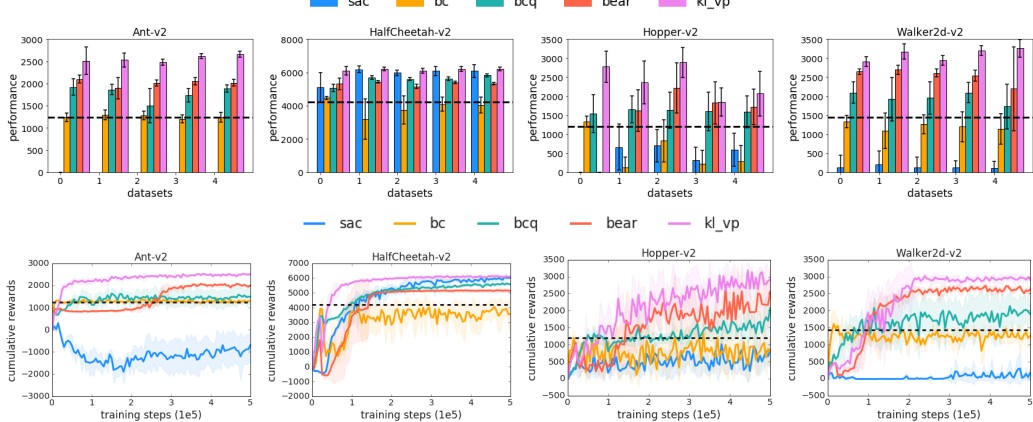

Figure 6: Comparing value penalty with KL divergence (kl_vp) to vanilla SAC, behavior cloning (bc), BCQ and BEAR. Bottom row shows sampled training curves with 1 out of the 5 datasets. See Appendix for training curves on all datasets.

## 4.6 HYPERPARAMETER SENSITIVITY

In our experiments, we find that many simple algorithmic designs achieve good performance under the framework of BRAC. For example, all of the 4 divergences we tried perform similarly well when used for regularization. In these experiments, we allowed for appropriate hyperparameter tuning over policy learning rate and regularization weight, as we initially found that not doing so can lead to premature and incorrect conclusions. [2] However, some design choices may be more robust to hyperparameters than others. To investigate this, we also analyzed the sensitivity to hyperparameters for all algorithmic variants (Appendix Figures 9 and 10). To summarize, we found that (i) MMD and KL Divergence are similar in terms of sensitivity to hyperparameters, (ii) using the dual form of divergences (e.g. KL dual, Wasserstein) appears to be more sensitive to hyperparameters, possibly because of the more complex training procedure (optimizing a minimax objective), and (iii) value penalty is slightly more sensitive to hyperparameters than policy regularization despite its more favorable performance under the best hyperparameters.

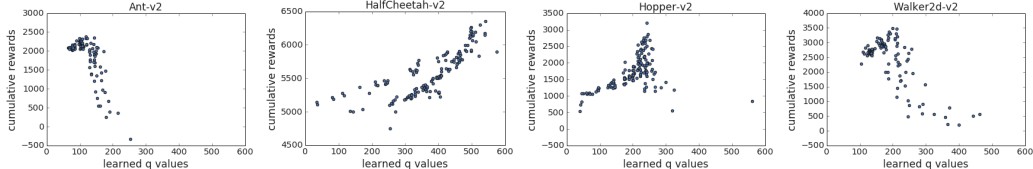

Figure 7: Correlation between learned Q-values and performance. x-axis is the average of learned $Q_\psi(s, a)$ over the last 500 training batches. y-axis is the average performance over the last 10 evaluation points. Each plot corresponds to a (environment, algorithm, dataset) tuple. Different points in each plot correspond to different hyperparameters and different random seeds.

Although we utilized hyperparameter searches in our results, in pure offline RL settings, testing on the real environment is infeasible. Thus, a natural question is how to select the best hyperparameter or the best learned policy among many without direct testing. As a preliminary attempt, we evaluated whether the Q-values learned during training can be used as a proxy for hyperparameter selection. Specifically, we look at the correlation between the average learned Q-values (in mini-batches) and the true performance. Figure 7 shows sampled visualizations of these Q-values. We find that the learned Q-values are not a good indicator of the performance, even when they are within a reasonable range (i.e., not diverging during training). A more formal direction for doing hyperparameter selection is to do off-policy evaluation. However, off-policy evaluation is an open research problem with limited

---

[2] For example, taking the optimal hyperparameters from one design choice and then applying them to a different design choice (e.g., MMD vs KL divergence) can lead to incorrect conclusions (specifically, that using KL is worse than using MMD, only because one transferred the hyperparameters used for MMD to KL).

success on complex continuous control tasks (see Liu et al. (2018); Nachum et al. (2019); Irpan et al. (2019) for recent attempts), we leave hyperparameter selection as future work and encourage more researchers to investigate this direction.

## 5 Conclusion

In this work, we introduced *behavior regularized actor critic* (BRAC), an algorithmic framework, which generalizes existing approaches to solve the offline RL problem by regularizing to the behavior policy. In our experiments, we showed that many sophisticated training techniques, such as weighted target Q-value ensembles and adaptive regularization coefficients are not necessary in order to achieve state-of-the-art performance. We found that the use of value penalty is slightly better than policy regularization, while many possible divergences (KL, MMD, Wasserstein) can achieve similar performance. Perhaps the most important differentiator in these offline settings is whether proper hyperparameters are used. Although some variants of BRAC are more robust to hyperparameters than others, every variant relies on a suitable set of hyperparameters to train well. Off-policy evaluation without interacting with the environment is a challenging open problem. While previous off-policy evaluation work focuses on reducing mean-squared-error to the expected return, in our problem, we only require a ranking of policies. This relaxation may allow novel solutions, and we encourage more researchers to investigate this direction in the pursuit of truly offline RL. Another potential direction is to look at the situations when the dataset is much smaller. Our preliminary observations on smaller datasets is that it is hard to get a hyperparameter that works consistently well on multiple runs with different random seeds. So we conjecture that smaller datasets may need either more careful hyperparameter search or (more interestingly) a better algorithm. We leave an extensive study of this setting to future work.

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

# A  ADDITIONAL EXPERIMENT RESULTS

## A.1  ADDITIONAL EXPERIMENT DETAILS

**Dataset collection**  For each environment, we collect five datasets: {no-noise, eps-0.1, eps-0.3, gauss-0.1, gauss-0.3} using a partially trained policy $\pi$. Each dataset contains 1 million transitions. Different datasets are collected with different injected noise, corresponding to different levels and strategies of exploration. The specific noise configurations are shown below:

- **no-noise** : The dataset is collected by purely executing the partially trained policy $\pi$ without adding noise.
- **eps-0.1**: We make an epsilon greedy policy $\pi'$ with 0.1 probability. That is, at each step, $\pi'$ has 0.1 probability to take a uniformly random action, otherwise takes the action sampled from $\pi$. The final dataset is a mixture of three parts: 40% transitions are collected by $\pi'$, 40% transitions are collected by purely executing $\pi$, the remaining 20% are collected by a random walk policy which takes a uniformly random action at every step. This mixture is motivated by that one may only want to perform exploration in only a portion of episodes when deploying a policy.
- **eps-0.3**: $\pi'$ is an epsilon greedy policy with 0.3 probability to take a random action. We do the same mixture as in eps-0.1.
- **gauss-0.1**: $\pi'$ is taken as adding an independent $\mathcal{N}(0, 0.1^2)$ Gaussian noise to each action sampled from $\pi$. We do the same mixture as in eps-0.1.
- **gauss-0.3**: $\pi'$ is taken as adding an independent $\mathcal{N}(0, 0.3^2)$ Gaussian noise to each action sampled from $\pi$. We do the same mixture as in eps-0.1.

**Hyperparameter search**  As we mentioned in main text, for each variant of BRAC and each environment, we do a grid search over policy learning rate and regularization strength. For policy learning rate, we search over six values: $\{3 \cdot 10^6, 1 \cdot 10^5, 3 \cdot 10^5, 0.0001, 0.0003, 0.001\}$. The regularization strength is controlled differently in different algorithms:

- In BCQ, we search for the perturbation range $\Phi \in \{0.005, 0.015, 0.05, 0.15, 0.5\}$. 0.05 is the reported value by its paper (Fujimoto et al., 2018a).
- In BEAR the regularization weight $\alpha$ is adaptively trained with dual gradient ascent based on a divergence constraint $\epsilon$ that is tuned as a hyperparameter. We search for $\epsilon \in \{0.015, 0.05, 0.15, 0.5, 1.5\}$. 0.05 is the reported value by its paper (Kumar et al., 2019).
- When MMD is used with a fixed $\alpha$, we search for $\alpha \in \{3, 10, 30, 100, 300\}$.
- When KL divergence is used with a fixed $\alpha$ (both KL and KL_dual), we search for $\alpha \in \{0.1, 0.3, 1.0, 3.0, 10.0\}$.
- When Wasserstein distance is used with a fixed $\alpha$, we search for $\alpha \in \{0.3, 1.0, 3.0, 10.0, 30.0\}$.

in BEAR the regularization weight is adaptively trained with dual gradient ascent based on a divergence constraint $\epsilon$ that is tuned as a hyperparameter;

In the simplest case, the regularization weight $\alpha$ is fixed; in BCQ the corresponding tuning is for the perturbation range $\Phi$. For each of these options, we search over five values (see Appendix for details). For existing algorithms such as BEAR and BCQ, the reported hyperparameters in their papers (Kumar et al., 2019; Fujimoto et al., 2018a) are included in this search range, We select the best hyperparameters according to the average performance over all five datasets.

**Implementation details**  All experiments are implemented with Tensorflow and executed on CPUs. For all function approximators, we use fully connected neural networks with RELU activations. For policy networks, we use $\tanh(\text{Gaussian})$ on outputs following BEAR (Kumar et al., 2019), except for BCQ where we follow their open sourced implementation. For BEAR and BCQ we follow the network sizes as in their papers. For other variants of BRAC, we shrink the policy networks from $(400, 300)$ to $(200, 200)$ and Q-networks from $(400, 300)$ to $(300, 300)$ for saving computation

time without losing performance. Q-function learning rate is always 0.001. As in other deep RL algorithms, we maintain source and target Q-functions with an update rate 0.005 per iteration. For MMD we use Laplacian kernels with bandwidth reported by Fujimoto et al. (2018a). For divergences in the dual form (both KL_dual and Wasserstein), we training a $(300, 300)$ fully connected network as the critic in the minimax objective. Gradient penalty (one sided version of the penalty in Gulrajani et al. (2017) with coefficient 5.0) is applied to both KL and Wasserstein dual training. In each training iteration, the dual critic is updated for 3 steps (which we find better than only 1 step) with learning rate 0.0001. We use Adam for all optimizers. Each agent is trained for 0.5 million steps with batch size 256 (except for BCQ we use 100 according their open sourced implementation). At test time we follow Kumar et al. (2019) and Fujimoto et al. (2018a) by sampling 10 actions from $\pi_\theta$ at each step and take the one with highest learned Q-value.

## A.2 VALUE PENALTY V.S. POLICY REGULARIZATION

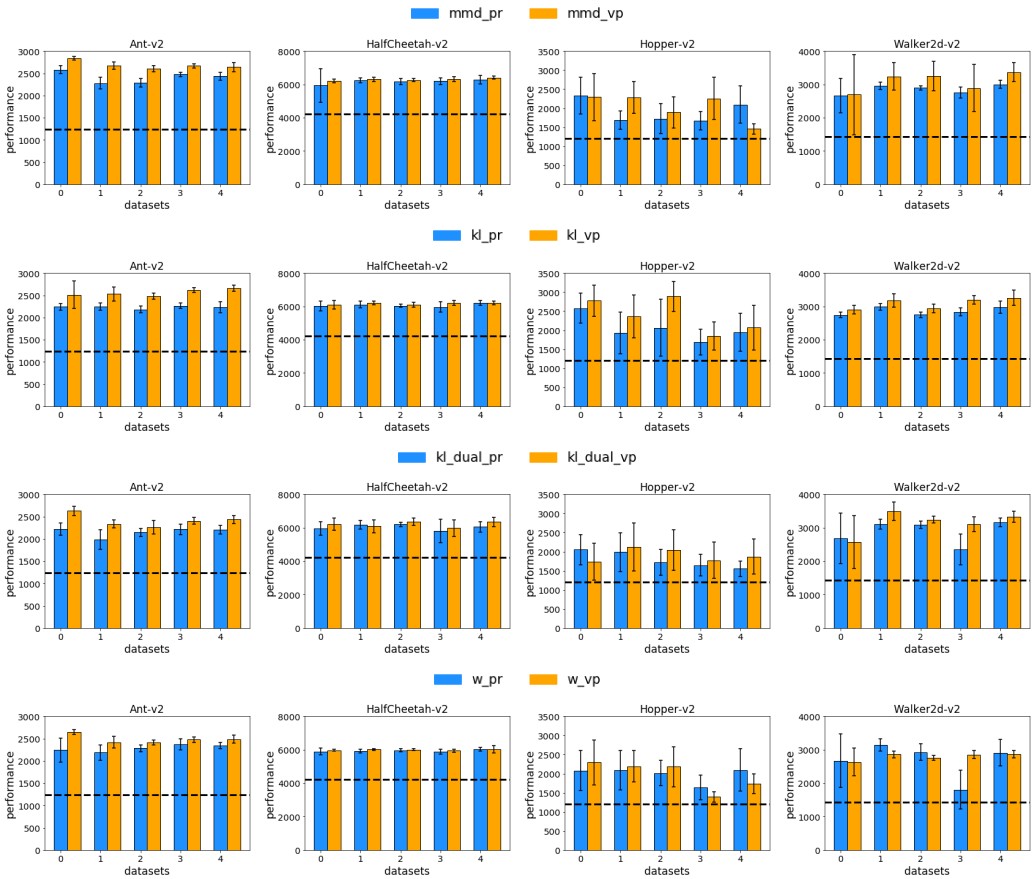

Figure 8: Comparing policy regularization (pr) v.s. value penalty (vp) with all four divergences. The use of value penalty is usually slightly better.

## A.3 FULL PERFORMANCE RESULTS UNDER DIFFERENT HYPERPARAMETERS

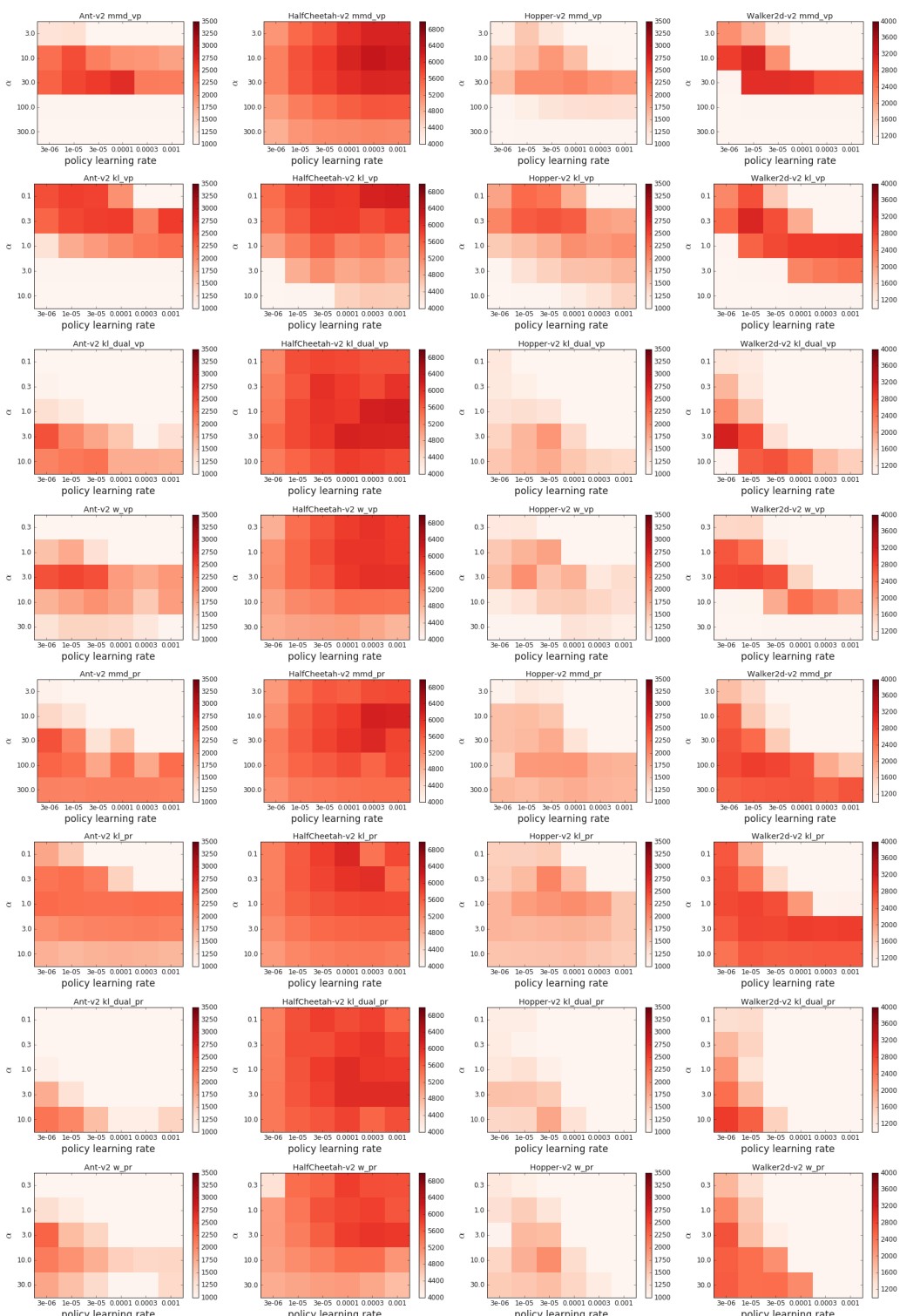

Figure 9: Visualization of performance under different hyperparameters. The performance is averaged over all five datasets.

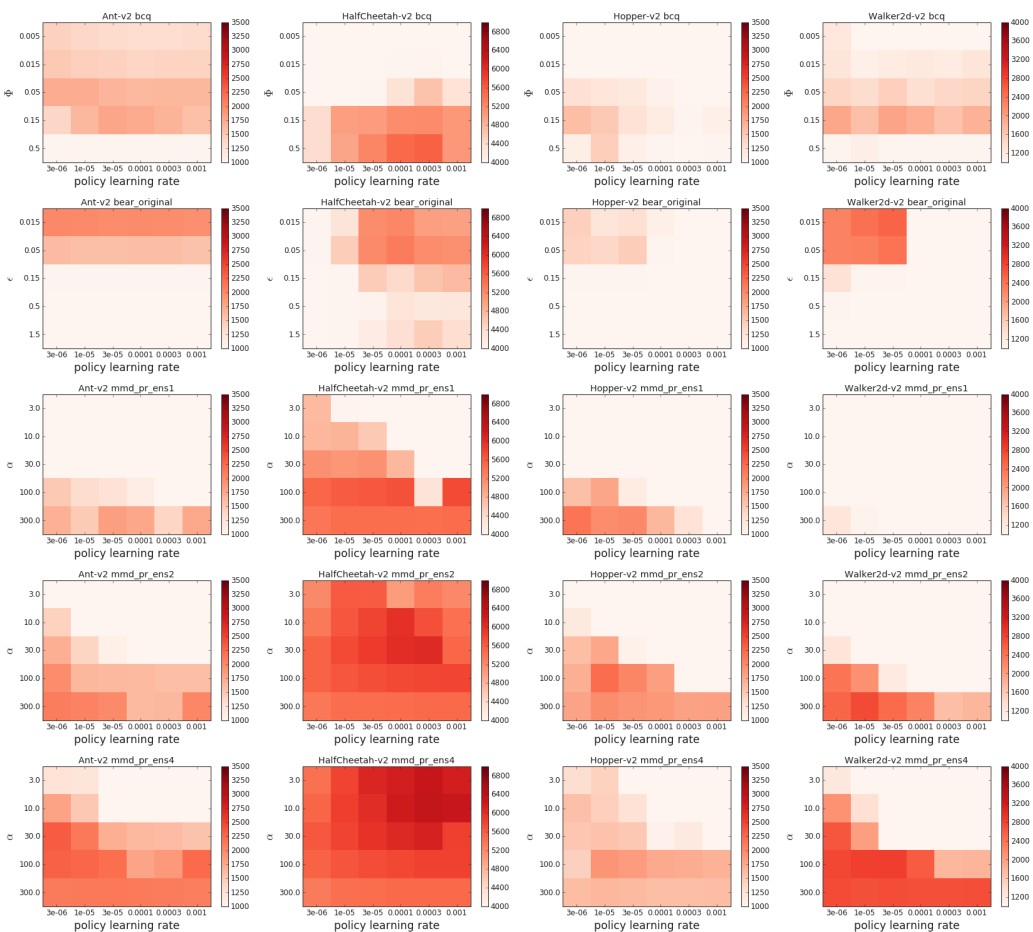

Figure 10: Visualization of performance under different hyperparameters.

## A.4 Additional training curves

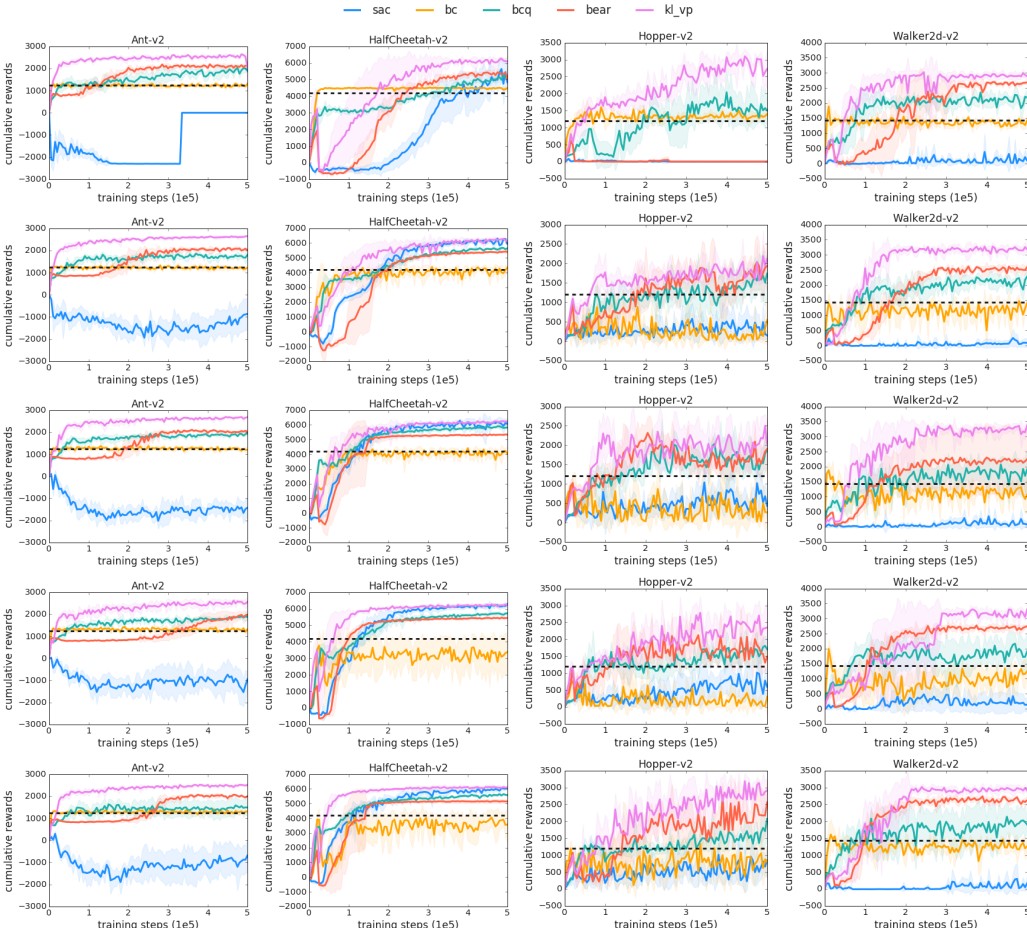

Figure 11: Training curves on all five datasets when comparing kl_vp to other baselines.

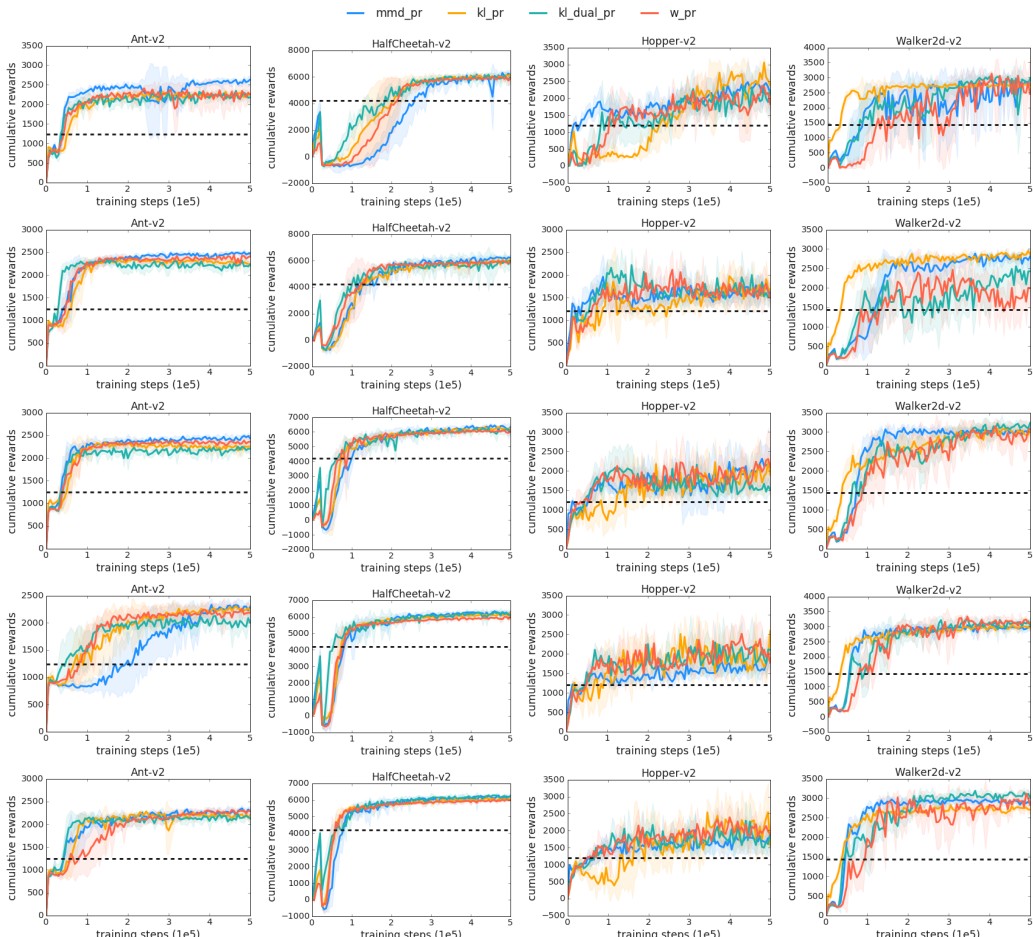

Figure 12: Training curves when comparing different divergences with policy regularization. All divergences perform similarly.

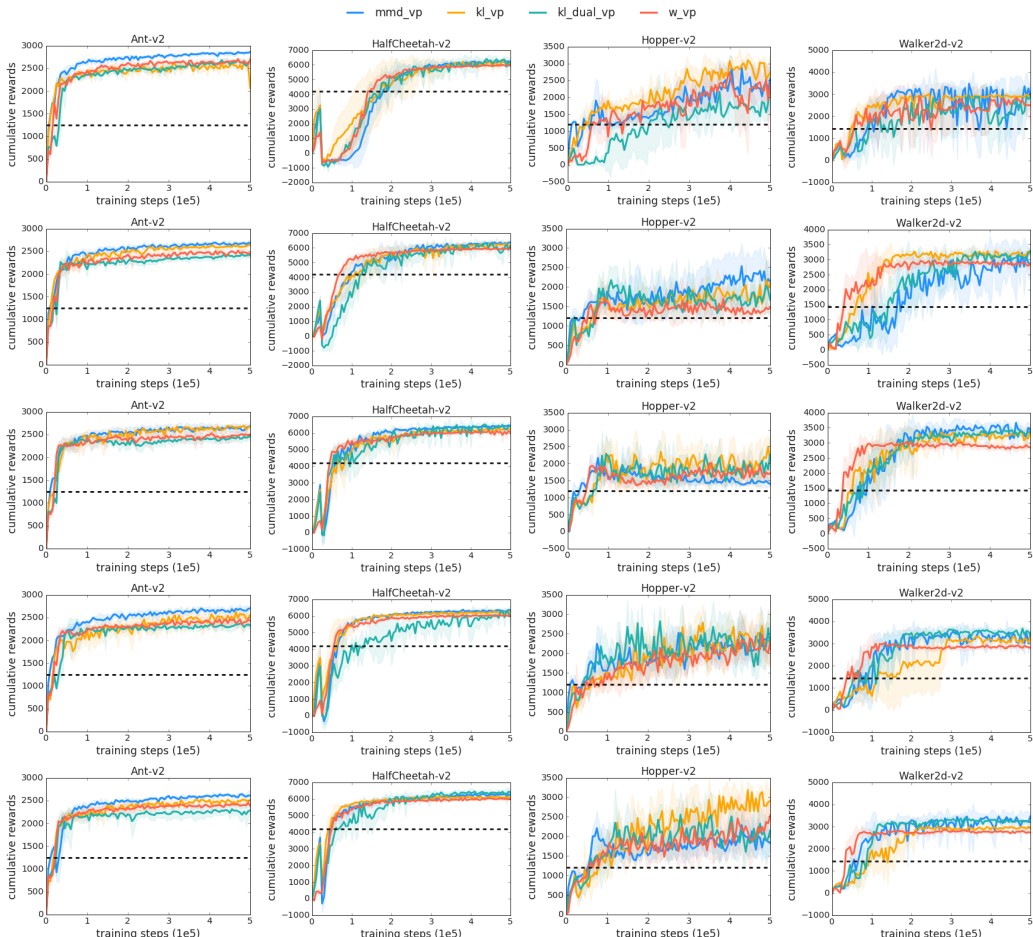

Figure 13: Training curves when comparing different divergences with value penalty. All divergences perform similarly.

## A.5 FULL PERFORMANCE RESULTS UNDER THE BEST HYPERPARAMETERS

| Environment: Ant-v2 | | | Partially trained policy: 1241 | | |
|---|---|---|---|---|---|
| dataset | no-noise | eps-0.1 | eps-0.3 | gauss-0.1 | gauss-0.3 |
| SAC | 0 | -1109 | -911 | -1071 | -1498 |
| BC | 1235 | 1300 | 1278 | 1203 | 1240 |
| BCQ | 1921 | 1864 | 1504 | 1731 | 1887 |
| BEAR | 2100 | 1897 | 2008 | 2054 | 2018 |
| MMD_vp | **2839** | **2672** | **2602** | **2667** | 2640 |
| KL_vp | 2514 | 2530 | 2484 | 2615 | **2661** |
| KL_dual_vp | 2626 | 2334 | 2256 | 2404 | 2433 |
| W_vp | 2646 | 2417 | 2409 | 2474 | 2487 |
| MMD_pr | 2583 | 2280 | 2285 | 2477 | 2435 |
| KL_pr | 2241 | 2247 | 2181 | 2263 | 2233 |
| KL_dual_pr | 2218 | 1984 | 2144 | 2215 | 2201 |
| W_pr | 2241 | 2186 | 2284 | 2365 | 2344 |

Table 1: Evaluation results with tuned hyperparameters. 0 performance means overflow encountered during training due to diverging Q-functions.

| Environment: HalfCheetah-v2 | | | Partially trained policy: 4206 | | |
|---|---|---|---|---|---|
| dataset | no-noise | eps-0.1 | eps-0.3 | gauss-0.1 | gauss-0.3 |
| SAC | 5093 | 6174 | 5978 | 6082 | 6090 |
| BC | 4465 | 3206 | 3751 | 4084 | 4033 |
| BCQ | 5064 | 5693 | 5588 | 5614 | 5837 |
| BEAR | 5325 | 5435 | 5149 | 5394 | 5329 |
| MMD_vp | **6207** | **6307** | **6263** | **6323** | **6400** |
| KL_vp | 6104 | 6212 | 6104 | 6219 | 6206 |
| KL_dual_vp | 6209 | 6087 | 6359 | 5972 | 6340 |
| W_vp | 5957 | 6014 | 6001 | 5939 | 6025 |
| MMD_pr | 5936 | 6242 | 6166 | 6200 | 6294 |
| KL_pr | 6032 | 6116 | 6035 | 5969 | 6219 |
| KL_dual_pr | 5944 | 6183 | 6207 | 5789 | 6050 |
| W_pr | 5897 | 5923 | 5970 | 5894 | 6031 |

Table 2: Evaluation results with tuned hyperparameters.

| Environment: Hopper-v2 | | | Partially trained policy: 1202 | | |
|---|---|---|---|---|---|
| dataset | no-noise | eps-0.1 | eps-0.3 | gauss-0.1 | gauss-0.3 |
| SAC | 0.2655 | 661.7 | 701 | 311.2 | 592.6 |
| BC | 1330 | 129.4 | 828.3 | 221.1 | 284.6 |
| BCQ | 1543 | 1652 | 1632 | 1599 | 1590 |
| BEAR | 0 | 1620 | 2213 | 1825 | 1720 |
| MMD_vp | 2291 | 2282 | 1892 | **2255** | 1458 |
| KL_vp | **2774** | **2360** | 2892 | 1851 | 2066 |
| KL_dual_vp | 1735 | 2121 | 2043 | 1770 | 1872 |
| W_vp | 2292 | 2187 | 2178 | 1390 | 1739 |
| MMD_pr | 2334 | 1688 | 1725 | 1666 | **2097** |
| KL_pr | 2574 | 1925 | 2064 | 1688 | 1947 |
| KL_dual_pr | 2053 | 1985 | 1719 | 1641 | 1551 |
| W_pr | 2080 | 2089 | 2015 | 1635 | **2097** |

Table 3: Evaluation results with tuned hyperparameters.

| Environment: Walker-v2 | | | Partially trained policy: 1439 | | |
|---|---|---|---|---|---|
| dataset | no-noise | eps-0.1 | eps-0.3 | gauss-0.1 | gauss-0.3 |
| SAC | 131.7 | 213.5 | 127.1 | 119.3 | 109.3 |
| BC | 1334 | 1092 | 1263 | 1199 | 1137 |
| BCQ | 2095 | 1921 | 1953 | 2094 | 1734 |
| BEAR | 2646 | 2695 | 2608 | 2539 | 2194 |
| MMD_vp | 2694 | 3241 | **3255** | 2893 | **3368** |
| KL_vp | **2907** | 3175 | 2942 | **3193** | 3261 |
| KL_dual_vp | 2575 | **3490** | 3236 | 3103 | 3333 |
| W_vp | 2635 | 2863 | 2758 | 2856 | 2862 |
| MMD_pr | 2670 | 2957 | 2897 | 2759 | 3004 |
| KL_pr | 2744 | 2990 | 2747 | 2837 | 2981 |
| KL_dual_pr | 2682 | 3109 | 3080 | 2357 | 3155 |
| W_pr | 2667 | 3140 | 2928 | 1804 | 2907 |

Table 4: Evaluation results with tuned hyperparameters.

