# OpenReview forum: "Behavior Regularized Offline Reinforcement Learning"
_ICLR.cc/2020/Conference — Reject_

### Official Review · AnonReviewer1 · 2019-10-21
**Official Blind Review #1**

**Rating:** 6

**Review:**

This paper proposes a unifying framework, BRAC, which summarizes the idea and evaluates the effectiveness of recently proposed offline reinforcement learning algorithms, specifically BEAR, BCQ, and KL control. The authors generalize existing offline RL approaches to an actor-critic algorithm that regularizes the learned policy such that it can stay close to the behavior policy. Based on prior work, the authors state that there are two variants of regularizations to the behavior policy, value penalty (vp) and policy regularization (pr) and three choices of divergence functions along with their sample estimate that measures the distance between the learned policy and the behavior policy. The paper empirically investigates the effectiveness of each regularization scheme as well as each divergence function and conclude that vp is slightly more effective than pr while all divergence functions have similar performances.

Overall, this paper could be an interesting summary of prior works in offline RL and provide some empirical insights on the effectiveness of each building block in the previous approaches, though it neither offers theoretical explanations nor proposes a new offline RL algorithm that outperforms the existing methods under the BRAC framework. My score would be increased given some technical insights or some promising results in a relatively novel offline RL algorithm in the author’s response.

Specifically, the paper does a thorough ablation study on BEAR, BCQ, and KL-control within the BRAC framework. While the experimental results demonstrate some interesting phenomenons such as combining vp and the primal form of KL divergence achieving the best performance and taking minimum over Q functions outperforming using a mix of maximum and minimum over the Q functions, I believe the paper would be greatly improved if the authors can provide a new offline RL method based on the BRAC that can achieve better performance than current approaches and is less incremental than simply combining vp and KL divergence.

Though BRAC summarizes offline RL methods in a neat way, it would be more technically sound if a general theoretical analysis/insights of offline RL algorithms can be offered in the paper, e.g. showing the reason that vp is outperforming pr through convergence analysis in the tabular case.

Minor comment:
Error bars should be added to the all the bar plots.

**UPDATE**
After reading the rebuttal, I think my concerns are addressed and thus I updated my rating.

**Experience Assessment:**

I have published one or two papers in this area.

**Review Assessment: Checking Correctness Of Derivations And Theory:**

N/A

**Review Assessment: Checking Correctness Of Experiments:**

I assessed the sensibility of the experiments.

**Review Assessment: Thoroughness In Paper Reading:**

I read the paper at least twice and used my best judgement in assessing the paper.

---

> ### Author Response · Authors · 2019-11-15
> **Response**
>
> We thank the reviewer for the valuable feedback. Responses to the reviewer’s comments are addressed below. We are also always happy to discuss further if the reviewer has additional concerns.
>
> The reviewer asks for a new offline RL method or theoretical analysis.
>
> -- While we agree that the paper would be improved with a novel, technical algorithm, the extensive empirical evaluation we present is a significant undertaking with the potential for large scientific impact. First, we believe that a systematic comparison of existing methodology can be an important piece of research contribution and at times a better contribution to the field than algorithmic novelty. Indeed, as a whole our empirical evaluations show that many previous algorithmic novelties (sophisticated Q-ensembles or divergences) are not necessary for good performance. We believe that in a research field it is important to periodically assess the utility of various algorithmic choices rather than always propose new ones, and this is what we strived to do in our paper. Previous work that has focused on empirical evaluations (e.g., “Deep RL that Matters” Henderson et. al. 2017, “A Large-Scale Study on Regularization and Normalization in GANs” Kurach et al. 2019, “Challenging Common Assumptions in the Unsupervised Learning of Disentangled Representations” Locatello et al. 2019) has had a significant impact on the community.
>
>
> Regarding theoretical analysis, there exist a number of previous works which look into some of the design choices we analyze:
> -- The use of min-Q-ensembles is known to reduce over-estimation of Q-values, thus leading to safer (more conservative) learned policies. (Hasselt 2010; also https://openreview.net/forum?id=Bkg0u3Etwr)
> -- When using KL regularization, the use of value-penalty can be interpreted as causal relative entropy, which as been extensively studied; see “A Theory of Regularized Markov Decision Processes” Geist, et al.
> However, these theoretical works still leave substantial gaps for a research practitioner in terms of explaining what makes a difference in empirical performance, especially when function approximation (e.g. neural networks) is involved. Our work can serve as important empirical evidence for future research that aims at closing the gap between theory and practice.
>
>
>
> “Error bars.”
>
> -- We have updated our paper with error bars added to our plots.
>
>
> Henderson, Peter, et al. "Deep reinforcement learning that matters." Thirty-Second AAAI Conference on Artificial Intelligence. 2018.
> Kurach, Karol, et al. "A Large-Scale Study on Regularization and Normalization in GANs." International Conference on Machine Learning. 2019.
> Locatello, Francesco, et al. "Challenging common assumptions in the unsupervised learning of disentangled representations." arXiv preprint arXiv:1811.12359 (2018).
> Hasselt, Hado V. "Double Q-learning." Advances in Neural Information Processing Systems. 2010.
> Geist, Matthieu, Bruno Scherrer, and Olivier Pietquin. "A Theory of Regularized Markov Decision Processes." arXiv preprint arXiv:1901.11275 (2019).

---

### Official Review · AnonReviewer3 · 2019-10-22
**Official Blind Review #3**

**Rating:** 6

**Review:**

The paper introduces a general framework for behavior regularized actor-critic methods, and empirically evaluates recent offline RL algorithms and different design choices.

Overall, the paper is well written and easy to follow.  I appreciate the authors for their careful empirical study. I am leaning to accept the paper because (1) the experimental design is rigorous and the results provide several insights into how to design a behavior regularized algorithm for offline RL.

There are some comments for the experiments.
1. Are the results significant (e.g. Figure 3 and 4)? Have you checked the error bars?
2. Missing numbers: trained_alpha in dataset 0 of Hopper-v2 in Figure 1 and SAC in dataset 0 of Hopper-v2 in Figure 6? Are they negative so not reported in the figure or just missing?
3. Do you think the conclusion will change if you use training datasets of different size (e.g. much less than 1 million)?

**Experience Assessment:**

I have published one or two papers in this area.

**Review Assessment: Checking Correctness Of Derivations And Theory:**

N/A

**Review Assessment: Checking Correctness Of Experiments:**

I assessed the sensibility of the experiments.

**Review Assessment: Thoroughness In Paper Reading:**

I read the paper at least twice and used my best judgement in assessing the paper.

---

> ### Author Response · Authors · 2019-11-15
> **Response**
>
> We thank the reviewer for the valuable feedback. Responses to the reviewer’s comments are addressed below. We are also always happy to discuss further if the reviewer has additional concerns.
>
> “Are the results significant (e.g. Figure 3 and 4)? Have you checked the error bars?”
>
> -- We have updated our paper with error bars added to our plots.
>
>
> “Missing numbers: trained_alpha in dataset 0 of Hopper-v2 in Figure 1 and SAC in dataset 0 of Hopper-v2 in Figure 6? Are they negative so not reported in the figure or just missing? ”
>
> -- When performance is negative we cutoff the bar at 0. For Hopper-2 in Figure 6, those results are numbers that are either negative or very close to 0 (e.g. 0.XX), thus not visible on the bar charts. We have updated the caption of Figure 1 to clarify this.
>
>
> “Do you think the conclusion will change if you use training datasets of different size (e.g. much less than 1 million)?”
>
> -- Considering training sets of much smaller size is an interesting direction. Our preliminary observations on smaller datasets suggest that the main difference from learning from a large dataset is that it is hard to get a hyperparameter that works consistently well on multiple runs (with different random seeds). So we conjecture that smaller datasets may need either more careful hyperparameter search or (more interestingly) a better algorithm. As our current experiments are already time consuming in terms of computation, due to lack of time, we leave an extensive study of this setting to future work. We have updated our paper to include this discussion in the conclusion section.

---

### Official Review · AnonReviewer2 · 2019-10-24
**Official Blind Review #2**

**Rating:** 6

**Review:**

This paper presents a framework for evaluating offline reinforcement learning (RL) algorithms.  Results from a  thorough series of experiments are presented which suggest that certain details of recently proposed RL methods are not necessary for achieving strong performance.  These results suggests that some of the complexity in RL design can be ignored.

I commend the authors for performing a valuable test and comparison of existing offline RL methodology.

This paper could be improved by providing more clear insight and intuition about the deeper meaning of these results regarding the "unnecessary" technical complexities.  Could the authors suggest why certain complexities are unnecessary?  Clearly the authors of those previous works thought they were needed.  To really help researchers design better algorithms, we need to be guided by some insight about not only what doesn't work but why it doesn't work.

Also, the paper could be improved by being more clear about the nature of the evaluations.  The authors provide extensive results; but it wasn't clear whether these were "apples-to-apples" comparisons with the previous results in the papers that proposed the "unnecessary" technical complexities.  For example, I didn't see the authors say that they reproduced the results of previous works, only that they tested previous methods in certain tests.  Does the BRAC framework reproduce the results for previous papers?  If so, this should be made more clear and stated prominently in the paper so that the reader knows that BRAC is, in this reproduction of previous results sense, reliable.  If not, how if the reader to know that the "unnecessary" technical complexities, are truly unneccessary?

Finally, the paper presents lots of results, but I did not see any mention of the statistical significance of these results.

Minor issue:  In the Conclusion Section, the authors say, "Unfortunately, off-policy ... is an challenging open problem."  Unfortunately?!  A challenging open problem is a good thing!  And I think the authors did a good job addressing a difficult problem.

**Experience Assessment:**

I have read many papers in this area.

**Review Assessment: Checking Correctness Of Derivations And Theory:**

I assessed the sensibility of the derivations and theory.

**Review Assessment: Checking Correctness Of Experiments:**

I assessed the sensibility of the experiments.

**Review Assessment: Thoroughness In Paper Reading:**

I read the paper at least twice and used my best judgement in assessing the paper.

---

> ### Author Response · Authors · 2019-11-15
> **Response**
>
> We thank the reviewer for the valuable feedback. Responses to the reviewer’s comments are addressed below. We are also always happy to discuss further if the reviewer has additional concerns.
>
> “This paper could be improved by providing more clear insight and intuition about the deeper meaning of these results regarding the "unnecessary" technical complexities… Clearly the authors of those previous works thought they were needed.”
>
> -- The intuition regarding the “unnecessary” technical complexities may be explained case-by-case.
>
>   For example, for “fixed v.s. adaptive regularization weights”, we conjecture that previous authors (BEAR) proposed adaptive regularization weights because it looks more robust (to e.g., reward scale). However, instead of tuning a fixed weight, adaptive weight requires another hyperparameter --- “epsilon-threshold” to be tuned, in addition to a new learning rate for updating the weight. In our experiments, we find that tuning the “threshold” is *not* easier than tuning the “fixed weight” (Appendix Figure 10). Furthermore, we find that the suggested threshold by BEAR’s paper is not playing an expected role, as we explained in Section 4.1 (the divergence never falls below the threshold, unlike adaptive regularization in SAC where the threshold can be reached). Therefore, we would suggest that using an “adaptive regularization weight” may be an unnecessary technical complexity.
>
>   Regarding “Q-ensemble”, our intuition is that more sophisticated ensemble (e.g. increase k, the number of Q-functions used for ensemble) will lead to better or more stable performance but requires more computation cost. The goal of our study here (Section 4.2) is to identify the minimum complexity needed in Q-ensemble to achieve near-best performance. We found that k=2 is necessary while k=4 only gives marginal improvement over k=2. We have updated Section 4.2 to better explain this.
>
>   In general, one difference between our work and previous papers is that we did a systematic extensive hyperparameter search on every algorithm/variant, which may lead to different conclusions on the necessity of some design choices compared to only doing limited hyperparameter search. In fact, in our initial experiments this was one of the first things we noticed: Taking the optimal hyperparameters from one design choice and then applying them to a different design choice (e.g., MMD vs KL divergence) can lead to incorrect conclusions (specifically, that using KL is worse than using MMD, only because one transferred the hyperparameters used for MMD to KL). This is another potential reason that previous works have claimed different conclusions. We have made a note of this in Section 4.6.
>
>
> “Does the BRAC framework reproduce the results for previous papers?”
>
> -- Although reproducing previous papers comes with the challenges of limited open source code and unreleased datasets, we performed a number of assessments to ensure that our implementation achieves similar performance as reported in the previous papers. Here is what we did for reproducing BEAR and BCQ respectively.
>
>   For BEAR, the code was not available at the time of our submission, so we closely followed their paper to reimplement their method. To check the performance of our implementation of BEAR, we follow the protocol BEAR’s paper used to collect the offline datasets with a partially trained policy. We confirmed that our partially trained policies are of similar performance to the ones reported in BEAR’s paper. Next, we confirmed that the policies learned by our implementation achieve similar or higher improvement over the partially trained policy as reported in BEAR’s paper in all environments.
>
>   For BCQ, our implementation exactly follows the open-source implementation (only differing in default initializations between Tensorflow and Pytorch). We confirmed the validity of our implementation by (a) first transferring our datasets to the format used by the open-source BCQ (b) then comparing the two implementations on the same datasets. We confirmed that our Tensorflow implementation and the open-source Pytorch implementation achieve near-identical results.
>
>   We will add the link of our open-source code to the final version of the paper so that in the future people can see and build upon the implementation we used in our comparison.
>
>
> “statistical significance of these results”
>
> -- We have updated our paper with error bars added to our plots.
>
>
> “Unfortunately”
>
> -- You’re right! We have updated text to be more optimistic about this opportunity.

---

### Decision · Program_Chairs · 2019-12-19

**Decision:**

Reject

**Comment:**

This paper is an empirical studies of methods to stabilize offline (ie, batch) RL methods where the dataset is available up front and not collected during learning. This can be an important setting in e.g. safety critical or production systems, where learned policies should not be applied on the real system until their performance and safety is verified. Since policies leave the area where training data is present, in such settings poor performance or divergence might result, unless divergence from the reference policy is regularized. This paper studies various methods to perform such regularization.

The reviewers are all very happy about the thoroughness of the empirical work. The work only studies existing methods (and combination thereof), so the novelty is limited by design. The paper was also considered well written and easy to follow. The results were very similar between the considered regularizers, which somehow limits the usefulness of the paper as practical guideline (although at least now we know that perhaps we do not need to spend a lot of time choosing the best between these). Bigger differences were observed between "value penalties" versus "policy regularization". This seems to correspond to theoretical observations by Neu et al (https://arxiv.org/abs/1705.07798, 2017), which is not cited in the manuscript. Although unpublished, I think that work is highly relevant for the current manuscript, and I'd strongly recommend the authors to consider its content. Some minor comments about the paper are given below.

On the balance, the strong point of the paper is the empirical thoroughness and clarity, whereas novelty, significance, and theoretical analysis are weaker points. Due to the high selectivity of ICLR, I unfortunately have to recommend rejection for this manuscript.

I have some minor comments about the contents of the paper:
- The manuscript contains the line:  "Under this definition, such a behavior policy πb is always well-defined even
if the dataset was collected by multiple, distinct behavior policies". Wouldn't simply defining the behavior as a mixture of the underlying behavior policies (when known) work equally well?
- The paper mentions several earlier works that regularize policies update using the KL from a reference policy (or to a reference policy). The paper of Peters is cited in this context, although there the constraint is actually on the KL divergence between state-action distributions, resulting in a different type of regularization.